# Beyond Infant Nutrition: Investigating the Long-Term Neurodevelopmental Impact of Breastfeeding

**DOI:** 10.3390/nu17162578

**Published:** 2025-08-08

**Authors:** Desislava Zhelyazkova, Maria Dzhogova, Simoneta Popova, Rouzha Pancheva

**Affiliations:** 1Department of Hygiene and Epidemiology, Faculty of Public Health, Medical University “Prof. Dr. Paraskev Stoyanov”, 9000 Varna, Bulgaria; simonetapopova@gmail.com; 2Medical Faculty, Medical University “Prof. Dr. Paraskev Stoyanov”, 9000 Varna, Bulgaria; mod2506mod@gmail.com

**Keywords:** breastfeeding duration, infant nutrition, early nutrition, neurodevelopment, NDT5

## Abstract

**Background/objectives**: Optimal infant nutrition, particularly exclusive breastfeeding for the first 6 months, is crucial for both immediate and long-term health. The early years of life are essential for brain development due of the rapid maturation of social, emotional, cognitive and motor capacities. While benefits of breastfeeding are well established, its long-term impact on neurodevelopment remains underexplored. This study investigates the relationship between breastfeeding duration and neurodevelopment outcomes at 5 years of age. **Methods**: This prospective cohort study followed 92 term-born infants in Varna, Bulgaria (2017–2024). Parents provided informed consent and completed questionnaires regarding demographic characteristics, feeding practices and atopic diseases. At 5 years of age, children were assessed using the Neurodevelopmental Test for Five-Year-Olds. **Results**: Feeding practices differed significantly across groups (*p* < 0.001), with exclusive breastfeeding more prevalent among children breastfed for longer. At 5 years, significant differences were observed in language development (*p* = 0.037) and behavioral outcomes (*p* = 0.001). A linear regression model for behavioral outcomes (F = 2.29, *p* = 0.011, R^2^ = 0.297) showed that breastfeeding for 6–12 months was associated with more favorable behavior (Estimate = −5.88, *p* = 0.026), compared to less than 6 months. In contrast, paternal secondary education (Estimate = 3.58, *p* = 0.048) compared to higher education and mixed ethnicity (Estimate = 12.55, *p* = 0.023) compared with Bulgarian ethnicity were associated with poorer behavioral outcomes (Estimate = 3.58, *p* = 0.048). **Conclusions**: Breastfeeding for 6 to 12 months may be associated with improved behavioral development, and to a lesser extent, language outcomes at age five. However, these domain-specific associations were not consistently supported across all statistical methods and should be interpreted with caution. Neurodevelopment is influenced by a complex interplay of nutritional, social and environmental factors. Longitudinal studies are needed to clarify the long-term effects of breastfeeding duration on neurodevelopment.

## 1. Introduction

Optimal infant nutrition—particularly exclusive breastfeeding for the first six months of life—plays a vital role in promoting both immediate and long-term health outcomes [1,2]. Global health guidelines recommend initiating breastfeeding within the first hour after birth and continuing for at least the first 2 years, alongside the introduction of safe and nutritionally adequate complementary foods. This recommendation is supported by a robust body of evidence demonstrating the significant health benefits of breastfeeding for both the infant and the mother [3].

For the child, breastfeeding reduces the risk of both acute and chronic illnesses, including respiratory and gastrointestinal infections, otitis media, sudden infant death syndrome (SIDS), necrotizing enterocolitis (especially in preterm infants), childhood leukemia, type 1 and type 2 diabetes, obesity and atopic dermatitis. It is also associated with improved neurodevelopmental outcomes, including modest but measurable increases in cognitive performance and IQ. Additionally, breastfeeding may support the development of the immune system and gut microbiota through bioactive components in human milk. These benefits are endorsed by the American Academy of Pediatrics and the Association of Women’s Health, Obstetric and Neonatal Nursing, both of which recommend exclusive breastfeeding for the first 6 months of life, followed by continued breastfeeding alongside complementary foods for at least 1 year or longer as mutually desired by the mother and infant [3,4,5,6,7].

Breastfeeding extends beyond providing nutrition to the infant; it is also a critical component of maternal recovery after childbirth. It facilitates the return to the pre-pregnancy physiological state by aiding in the resolution of the cardiovascular and metabolic changes that occur during pregnancy. Moreover, breastfeeding offers substantial long-term health benefits for mothers, including a reduced risk of breast and ovarian cancers, type 2 diabetes, hypertension and cardiovascular disease. It may also support postpartum weight loss and contribute to extended birth spacing through lactational amenorrhea. The hormonal responses associated with breastfeeding—particularly the release of oxytocin and prolactin—foster maternal–infant bonding and are associated with a decreased risk of postpartum depression [8,9,10].

Breastfeeding provides extensive and well-documented health benefits for both mother and child, along with broader societal and economic advantages through reduced healthcare costs and improved population health outcomes [3,11].

Early life nutrition plays a vital role in shaping the architecture and function of the developing brain, and the “first 1000 days”—from conception to a child’s second birthday—representing a critical window of sensitivity. Nutritional deficiencies or imbalances during this period can irreversibly impair key neurodevelopmental processes such as neuronal proliferation, myelination and synaptogenesis, leading to measurable deficits in cognitive, motor and behavioral outcomes. Sufficient intake of protein, iron, zinc, iodine, choline, folate, vitamins A, D, B6, B12 and long-chain polyunsaturated fatty acids is particularly essential during this stage [12,13,14]. Human milk provides an optimal nutrient composition along with bioactive molecules that are difficult to replicate in formula. Breastfed infants consistently exhibit modest but significant cognitive advantages compared to their formula-fed counterparts. Neuroimaging and EEG studies have shown that these children experience accelerated brain maturation and enhanced myelination [15,16,17].

The long-term impact of early nutrition, particularly breastfeeding, is supported by the concept of “programming,” which suggests that stimuli or insults during critical developmental periods can lead to lasting effects on an organism’s physiological and metabolic functions [18]. Emerging evidence from randomized intervention trials demonstrates that the benefits of early nutrition on neurodevelopment extends into adolescence and adulthood, supporting enhanced executive functioning, mental growth and intellectual ability [18,19].

In light of this evidence, the World Health Organization recommends exclusive breastfeeding for the first 6 months, followed by continued breastfeeding alongside appropriate complementary foods up to 2 years or beyond. This practice is associated with improved cognitive performance and reduced risk of neurodevelopmental impairments, with particularly pronounced benefits observed in resource-limited settings [17].

Neurodevelopment refers to the complex, time-sensitive processes through which a child acquires its social, emotional, cognitive and motor skills [1,3]. These developmental trajectories accelerate during the “first 1000 days,” making early nutrition and related lifestyle factors critical determinants of lifelong health and disease risk [1]. Breastfeeding is widely recognized as the gold standard for infant feeding, associated with lower rates of infection, obesity and metabolic disorders, as well as modest but measurable improvements in cognitive performance from infancy into adulthood. Neuroimaging studies have linked human-milk exposure to increased white- and grey-matter volumes, accelerated myelination, and more mature EEG patterns. However, the precise magnitude and persistence of these benefits —particularly in relation to feeding mode, exclusivity and duration—are not yet fully understood [4].

Evidence on the long-term neurodevelopmental impact of breastfeeding in Eastern European populations remains limited, with virtually no studies utilizing culturally validated assessment tools such as the Bulgaria-validated Neurodevelopmental Test for Five-Year-Olds (NDT5) [20]. This gap restricts the external validity of existing global recommendations and leaves clinicians in the region without population-specific data to guide nutritional and family counselling. To address this shortcoming, the present prospective cohort study investigates the intersection between the duration of breastfeeding and neurodevelopmental outcomes at 5 years of age in a birth cohort from Varna, Bulgaria. Potential confounding factors influencing early feeding practices are also examined to better understand how breastfeeding duration may shape the trajectory of neurological growth and development.

## 2. Materials and Methods

### 2.1. Study Design and Participants

This prospective cohort study followed 92 term-born infants in Varna, Bulgaria, from birth to 5 years of age between 2017 and 2024. Data were collected through parental reports and medical records. Participants were enrolled at birth. Inclusion criteria comprised full-term birth (≥37 weeks gestation), absence of major congenital anomalies and no history of significant perinatal complications or NICU admission in order to minimize variability due to early neonatal morbidities. Exclusion criteria included preterm birth, severe neonatal complications (e.g., hypoxic-ischemic encephalopathy, neonatal sepsis) or parental refusal to participate in follow-up assessments.

Of the 158 families approached at birth, 111 consented to participate (70.3% participation rate). Follow-up assessments were conducted at 1, 2 and 5 years of age. At the 5-year follow-up, 92 children (82.9% of those enrolled) remained in the study and were successfully assessed. Parents completed a structured questionnaire addressing demographic characteristics, feeding practices and personal or family history of atopic diseases. A physician subsequently performed a standardized neurodevelopmental assessment on each child.

### 2.2. Ethical Considerations

The study protocol was approved by the Ethics Committee of the Medical University under protocol numbers 60 (23 February 2017), 115 (31 March 2022) and 121 (6 October 2022).

Written informed consent was obtained from all parents or legal guardians prior to participation. Although formal assent was not required due to the young age of participants, each child was verbally asked whether they wished to engage in the planned activities and participation proceeded only with their clear verbal agreement and comfort. All procedures were conducted in accordance with pediatric ethical standards and national child protection guidelines.

### 2.3. Demographic and Environmental Variables

Collected demographic and environmental data included parental education, ethnicity, mode of delivery, region of residence, nursery and kindergarten attendance and parental smoking habits. Pet exposure was recorded at two time points—at birth and at 5 years of age—using three categories: no pets, pets kept outside the home and pets kept inside the home. All the data were collected retrospectively via parental report, under the guidance of a physician or a lactation specialist. These professionals applied standardized questionnaire strategies designed to minimize recall bias, which remains a potential limitation of the study.

### 2.4. Assessment of Infant Feeding Practices

Information on feeding practices was collected using a structured questionnaire administered with the support of lactation and infant nutrition specialists. Data included the duration of feeding, exclusivity and the type of formula used. Infant feeding practices were categorized into exclusive breastfeeding, mixed feeding (breast milk and formula) and exclusive formula feeding. Based on breastfeeding duration, participants were divided into three groups: Group A, breastfed for 6 months or less (short duration); Group B, breastfed between 6 and 12 months (medium duration); and Group C, breastfed for more than 12 months (long duration). For ease of interpretation, these groups are referred to throughout the manuscript as short, medium and long breastfeeding duration.

Feeding history was collected retrospectively through parental reports. Although trained lactation consultants facilitated structured responses to reduce recall bias, this remains a potential limitation of the study.

### 2.5. Neurodevelopmental Assessments

At 5 years of age, neurodevelopmental outcomes were evaluated using the Neurodevelopmental Test for Five-Year-Olds (NDT5) [20], a structured and age-appropriate assessment tool validated for use in Bulgaria. The NDT5 examines five core domains: motor development, speech and language abilities, articulation, nonverbal intelligence and behavioral characteristics. Each domain produces an individual numeric score, which are then summed to generate a total neurodevelopmental score. Higher total scores reflect poorer overall neurodevelopmental functioning, with a cumulative score above 64 points indicating a neurodevelopmental deficit.

The NDT5 demonstrates strong psychometric properties, including internal consistency (Cronbach’s alpha = 0.86) and test–retest reliability (ICC = 0.88). Although not internationally standardized, the tool is widely applied in pediatric and developmental clinical practice across Bulgaria. A Appendix A to the tool and its validation study (in Bulgarian) is provided with this manuscript.

### 2.6. Statistical Analysis

Statistical analyses were performed using Welch’s One-Way ANOVA to evaluate differences in neurodevelopmental outcomes among the breastfeeding duration groups, accounting for potential inequality of variances. Partial correlation analyses were conducted to assess associations between breastfeeding duration and neurodevelopmental outcomes, adjusting for key confounders, including parental age and education, maternal smoking during pregnancy, household exposure to smoking or vaping, feeding type, mode of birth, ethnicity, region of residence and pet exposure.

Normality of distribution was assessed using the Shapiro–Wilk test. Given that some outcome variables, such as language and behavioral scores, deviated from normality, the use of Welch’s ANOVA and non-parametric correlation methods was deemed appropriate.

Univariate and multivariate linear regression models were used to identify predictors of behavioral and cognitive outcomes, adjusting for relevant demographic, nutritional and environmental variables. The selection of covariates was informed by existing literature on early childhood neurodevelopmental outcomes. Variables such as parental education, ethnicity, smoking exposure and region of residence were included due to their known or hypothesized influence on developmental trajectories through socioeconomic, behavioral and environmental mechanisms [21,22,23,24].

Variables were retained in the multivariate model if they showed significance in univariate analysis or improved model fit based on Akaike Information Criterion (AIC) and Bayesian Information Criterion (BIC). Multicollinearity was assessed using variance inflation factors (VIFs); all included variables had VIFs below 2.5, indicating an acceptable level of collinearity. A two-tailed significance level of *p* < 0.05 was applied to all statistical tests. Sample size justification: a post hoc power analysis was conducted using G*Power 3.1 to determine whether the sample size was adequate for detecting effects in the multivariate regression model examining behavioral development outcomes. The final model included 14 predictors. Assuming a medium effect size (f^2^ = 0.429, corresponding to R^2^ = 0.297), α = 0.05 and power = 0.80, the minimum required sample size was estimated at 53 participants. With 92 children assessed at the 5-year-follow-up, the sample size was sufficient to detect medium-to-large effects. However, subgroup comparisons, particularly for the medium breastfeeding duration group (Group B, n = 10), remain underpowered for detecting small effects and should be interpreted with caution.

## 3. Results

### 3.1. Demographic and Baseline Characteristics

Table 1 presents an overview of the study sample characteristics by breastfeeding duration. Maternal age was significantly higher in the long breastfeeding duration group (Group C: >12 months; M = 38.2 years) compared to the short duration group (Group A: ≤ 6 months; M = 35 years; *p* = 0.005 **). Paternal education also differed significantly across groups, with a higher proportion of fathers in long breastfeeding duration group holding a master’s degree (77.8%) compared to the short duration group (43.8%; *p* = 0.033 *). Infant feeding practices varied markedly by group (*p* < 0.001 ***); exclusive breastfeeding was most prevalent in the medium (Group B) and long (Group C) duration groups. Moreover, mixed and formula feeding were more common in the short duration group (Group A). Smoking behaviors followed a notable trend: maternal smoking during pregnancy was more frequent in Group A (short duration) (21.9%) and infrequent in Group C (long duration) (5.6%), though this difference did not reach statistical significance (*p* = 0.085).

### 3.2. Neurodevelopmental Outcomes by Breastfeeding Duration

The analysis revealed several trends in neurodevelopmental outcomes across breastfeeding duration groups, where lower scores indicate more favorable neurodevelopmental functioning (Table 2).

A significant difference in **language development** was observed among the groups (***p* = 0.037 ***). Children in the long breastfeeding duration group (Group C) demonstrated the lowest mean score (5.56), indicating the best language outcomes. This was followed by the medium duration group (Group B; mean = 11.60), while the short duration group (Group A) had the highest (i.e., poorest) mean score (12.91). These results suggest a positive association between longer breastfeeding duration and improved language development.

**Behavioral outcomes** also differed significantly between groups (***p* = 0.001 *****). The medium breastfeeding duration group exhibited the most favorable outcomes (mean score 0.90), followed by the long duration group (mean = 3.78). The short duration group again had the highest (i.e., poorest) mean score (5.67), indicating greater behavioral difficulties among those children breastfed for shorter durations.

Although the differences in the **total developmental score** were not statistically significant (*p* = 0.313), both the medium and long breastfeeding duration groups had notably lower mean scores (32.60 and 32.61, respectively) compared to the short duration group (47.94. This trend may indicate an overall association between longer breastfeeding duration and more favorable global neurodevelopmental outcomes, albeit without reaching statistical significance.

### 3.3. Partial Correlation Results at 5 Years

Partial correlation analyses were performed to examine the associations between breastfeeding duration and neurodevelopmental outcomes across various domains at 5 years of age, while statistically controlling for key confounding variables. These included maternal and paternal education, parental age, maternal smoking during pregnancy, household exposure to smoking or vaping, feeding type, mode of delivery and ethnicity. When breastfeeding duration was categorized into three groups (short, medium and long), no statistically significant associations were identified between breastfeeding duration and any of the developmental outcomes. All *p*-values exceeded 0.05, indicating the absence of independent linear relationships after adjustment for covariates. The detailed results are presented in Appendix A.

This lack of significance in partial correlation analyses contrasts with the group-level differences observed in the ANOVA. This discrepancy suggests that the relationship between breastfeeding duration and developmental outcomes may not be adequately captured by linear correlation alone. Instead, complex, non-linear or interaction effects, potentially involving environmental or familial factors, may underlie the observed associations—effects that are better elucidated through multivariate regression modeling.

### 3.4. Predictors of Developmental Outcomes at 5 Years: A Multivariate Linear Regression Approach

To investigate the factors associated with overall developmental outcomes at 5 years of age, we conducted a series of linear regression analyses using multivariate models. The analysis began with univariate testing to assess the individual effects of each independent variable on the total neurodevelopmental score.

Subsequently, a stepwise selection method was applied to iteratively include or exclude variables based on their statistical relevance. The final multivariate model was selected based on indicators of optimal model fit, including the lowest Akaike Information Criterion (AIC) and Bayesian Information Criterion (BIC) values, as well as the highest adjusted R^2^ and overall statistical significance.

#### 3.4.1. Linear Regression Model for Total Developmental Score

A multivariate linear regression model was conducted to identify predictors of total neurodevelopmental outcomes at 5 years of age (Table 3 and Table 4). The final model accounted for 27.3% of the variance in total developmental scores (R^2^ = 0.273, *p* = 0.003).

Region of residence emerged as a significant predictor (β = 54.02, SE = 22.04, *p* = 0.016), suggesting that children from different residential areas demonstrated distinct developmental trajectories. Ethnicity showed a significant association with developmental outcomes, especially in comparisons where the mixed ethnicity group was contrasted with the reference group (β = 98.70, SE = 36.94, *p* = 0.009).

Paternal education level demonstrated a trend toward significance (β = 19.50, SE = 11.34, *p* = 0.089), indicating a possible positive association between higher paternal education and better developmental outcomes. Additionally, a significant negative association was observed between paternal smoking or vaping at the child’s age of 5 and total developmental scores (β = −24.29, SE = 11.92, *p* = 0.045).

While breastfeeding duration did not independently predict total developmental scores in this model, it contributed to overall model fit and may interact with other variables in shaping developmental trajectories.

#### 3.4.2. Linear Regression Model for Behavioral Developmental Score

A linear regression model was developed to identify predictors of behavioral outcomes at 5 years of age (Table 5 and Table 6). The overall model was statistically significant (F = 2.29, ***p* = 0.011 ****), explaining approximately 29.7% of the variance in behavioral development (R^2^ = 0.297).

Several predictors demonstrated statistically significant associations with behavioral development at 5 years. Breastfeeding for 6–12 months (medium duration) was associated with significantly better behavioral outcomes compared to breastfeeding for less than 6 months (Estimate = −5.88, *p* = 0.026). In contrast, having a father with only secondary education was linked to poorer behavioral outcomes relative to children whose fathers had higher education (Estimate = 3.58, *p* = 0.048). Children of mixed ethnicity exhibited significantly poorer behavioral scores compared to those of Bulgarian ethnicity (Estimate = 12.55, *p* = 0.023). Lastly, paternal smoking or vaping when the child was 5 years old was associated with better behavioral scores (Estimate = −3.78, *p* = 0.038).

#### 3.4.3. Linear Regression Model for Language Development

A linear regression model was constructed to identify predictors of language development scores at 5 years of age (Table 7 and Table 8). The overall model was statistically significant (F = 2.40, *p* = 0.007), accounting for 32.5% of the variance in language development outcomes (R^2^ = 0.325).

Two predictors emerged as statistically significant. Residence location was strongly associated with language outcomes: children residing in rural areas demonstrated significantly poorer language development compared to those in urban settings (Estimate = 20.24, *p* = 0.004). Similarly, children of mixed ethnicity had significantly worse language development scores compared to children of Bulgarian ethnicity (Estimate = 22.11, *p* = 0.045).

No other variables—including breastfeeding duration, parental education, parental smoking or vaping, family history of atopic dermatitis, maternal allergy or childhood atopic dermatitis—were significantly associated with language outcomes in this model.

#### 3.4.4. Overall Model Performance for the Other Tested Areas of Development

The same multivariate linear regression approach was applied to the remaining neurodevelopmental domains. The model predicting motor development accounted for the lowest proportion of variance (R^2^ = 0.142, *p* = 0.110), indicating limited explanatory power. In contrast, the models for articulation and nonverbal intelligence explained a greater proportion of variance, with R^2^ values of 0.227 (*p* < 0.001) and 0.149 (*p* = 0.007), respectively. Each model captured distinct aspects of child development, underscoring the domain-specific nature of early influences on neurodevelopmental outcomes.

## 4. Discussion

The present study’s findings corroborate existing evidence linking longer breastfeeding duration with improved neurodevelopmental outcomes at 5 years of age. Previous research, including studies by Plunkett et al. and Strøm et al., has demonstrated that breastfeeding for at least 1 month is associated with higher IQ scores, with each additional month contributing to a decreased risk of low IQ [25,26]. Notably, Strøm et al. observed that this relationship did not follow a linear dose–response pattern beyond the initial month [25]. In addition, Angelsen et al. reported a cognitive benefit among children breastfed for 6 months or more compared to those breastfed for less than 3 months, even after adjusting for parental IQ and educational background [27].

A multitude of confounding factors influencing neurodevelopment have been identified in the literature. These encompass genetic influences; socioeconomic determinants such as parental cognitive abilities and household income; and prenatal and perinatal exposures including maternal smoking, alcohol consumption, nutritional deficiencies, preeclampsia and infections. Additional factors with documented impact include maternal and perinatal stress, parental age, birth weight, gestational age and perinatal complications like hypoxia and respiratory distress [21,22,23,24].

Interestingly, Girard et al. reported that while breastfeeding conferred early cognitive benefits, these differences were no longer statistically significant by age five after adjusting for confounders [28]. This finding underscores the complexity of the relationship between breastfeeding and neurodevelopment, highlighting the substantial influence of contextual factors such as family environment, parental education and socioeconomic status. As Victora et al. (2016) emphasized, families with higher socioeconomic status typically have greater access to resources that support both breastfeeding and cognitive development, including maternal mental health services and enriched learning environments [10,29]. These confounding factors must be carefully considered when interpreting study outcomes.

Recent large-scale cohort studies further illustrate this complexity. For example, data from the UK Avon Longitudinal Study of Parents and Children (ALSPAC) demonstrated that children breastfed for at least 6 months exhibited significantly higher IQ scores at ages 8 and 15, even after controlling for socioeconomic and demographic confounders, with estimated IQ gains of 4.1 and 5.1 points, respectively [30]. In contrast, findings from the U.S. Early Childhood Longitudinal Study (ECLS) suggest that the positive association between breastfeeding and early cognitive performance may largely reflect family-level confounders. Within-sibling comparisons revealed no significant association between breastfeeding duration and cognitive outcomes [31]. Collectively, these results highlight the necessity of rigorously controlling for socioeconomic, familial and cultural variables and suggest that the relationship between breastfeeding and neurodevelopment is both context-dependent and domain-specific.

In our sample, significant differences were identified in maternal age, paternal education, and infant feeding practices across breastfeeding duration groups, whereas other characteristics—including sex distribution, residence, mode of birth and maternal education—did not differ significantly. Although variations in smoking behaviors were observed, these differences did not reach statistical significance. The significant differences in maternal age (*p* = 0.005) and paternal education (*p* = 0.033) suggest that families with older mothers and more highly educated fathers may be more likely to receive guidance and support conducive to longer breastfeeding durations.

Nursery and kindergarten attendance, as well as pet exposure, were initially included as covariates based on literature indicating modest positive effects on neurodevelopment [32,33]. However, these factors were not significant predictors within our regression models. Feeding practices were closely linked to breastfeeding duration, with exclusive breastfeeding more commonly sustained beyond 12 months, whereas mixed feeding was predominantly observed among infants with shorter breastfeeding durations (<6 months). These findings highlight the critical role of early feeding choices in influencing the persistence of breastfeeding over time.

Our literature review indicates that breastfeeding indicators in Bulgaria remain substantially below international recommendations and lag behind many countries in the WHO European Region. National surveys conducted in 2007 and 2014 demonstrate that while the vast majority of Bulgarian infants are breastfed in the first days of life (90.7% in 2007 and 86.3% in 2014), the rate of early initiation of breastfeeding—defined as within the first hour after birth and strongly recommended by the WHO—was exceedingly low, at 1.8% and 9.9%, respectively. These findings align with WHO regional data from 1998 to 2013, which reported early initiation rates as low as 5% in Bulgaria, the lowest among the 21 countries assessed. Exclusive breastfeeding (EBF) rates are particularly limited; in Bulgaria, only 12.5% of infants under 6 months were exclusively breastfed in 2007, increasing modestly to 21.7% in 2014. WHO regional comparisons reveal even lower EBF prevalence, with approximately 6% of infants exclusively breastfed under 4 months and 2% under 6 months, again among the lowest rates recorded in the region. By contrast, countries such as Kyrgyzstan, Georgia and Croatia report EBF rates exceeding 50% for infants under 6 months. Despite relatively high initial breastfeeding rates, the overall duration of breastfeeding in Bulgaria remains short and sustained breastfeeding beyond infancy is rarely documented due to limited national research. These patterns reflect suboptimal adherence to WHO and UNICEF guidelines and emphasize the urgent need for updated national policies and enhanced maternal education on infant nutrition and breastfeeding best practices [34,35].

Although we observed some differences across breastfeeding duration groups, our analyses did not reveal a consistent or strong association between breastfeeding duration and most developmental outcomes at age five. Welch’s ANOVA indicated statistically significant differences in language development (*p* = 0.037 *) and behavioral scores (*p* = 0.001 ***), while no significant differences were found for motor development (*p* = 0.195), articulation (*p* = 0.980) or nonverbal intelligence (*p* = 0.660). Partial correlation analyses, which adjusted for key confounders, showed no significant associations between breastfeeding duration and any developmental outcomes. These results suggest that breastfeeding duration, when considered independently, may not be a robust predictor of neurodevelopmental outcomes after accounting for other influential factors.

However, our multivariate regression analyses revealed a more nuanced picture. Breastfeeding for 6–12 months was significantly associated with improved behavioral scores (*p* = 0.026 *), consistent with the group differences observed in ANOVA. Yet this association was not supported by the partial correlation analysis (*p* = 0.327), indicating that the relationship between breastfeeding and behavioral development is likely complex and not solely attributable to breastfeeding duration. Instead, breastfeeding may interact with multiple environmental and familial factors to influence neurodevelopment.

Overall, our findings suggest that breastfeeding may contribute to certain domains of child development—particularly behavioral outcomes—but not in isolation. These results should be interpreted cautiously and further research using larger and more diverse cohorts is warranted to elucidate the interplay between early feeding practices and other life factors in shaping long-term developmental trajectories.

The study’s findings are consistent with previous literature emphasizing the developmental benefits of breastfeeding [27,36,37]. However, the lack of statistically significant effects for breastfeeding beyond 12 months warrants further investigation, as this may reflect a plateau in benefit or the influence of unmeasured cofounding factors. While our study primarily focused on breastfeeding duration, feeding practices themselves may independently impact neurodevelopment. Although duration and type of feeding were correlated in our cohort, future research should evaluate these variables concurrently to clarify whether feeding practices serve as stronger or complementary predictors of developmental outcomes.

Emerging evidence from diverse populations supports a dose–response relationship between breastfeeding and early neurodevelopment. For instance, Zheng et al. (2024), in a cohort of Chinese children aged 2–3 years, reported that breastfeeding for 7–12 months was significantly associated with higher neurodevelopmental scores across motor and language domains, compared to children who were never breastfed [38]. Saigh (2025) found that breastfeeding for 12 to 24 months was linked to a substantially reduced risk of autism spectrum disorders, potentially mediated by immune modulation and gut–brain axis interactions [39]. In one recent large cohort study, Goldshtein et al. (2025) analyzed data from over 570,000 Israeli children and demonstrated that breastfeeding for 6 months or longer—whether exclusive or not—was associated with lower odds of developmental delays [40]. Notably, these protective effects remained significant in sibling-controlled analyses, underscoring the robustness of the association. Collectively, these findings highlight the critical role of both breastfeeding duration and exclusivity, suggesting consistent neurodevelopmental benefits across diverse cultural and socioeconomic settings.

An important and unexpected finding in our study was that children of mixed ethnicity exhibited significantly lower developmental scores compared to their peers of Bulgarian ethnicity. This observation raises critical considerations that require careful and nuanced interpretation. One potential explanation is the presence of cultural bias within the neurodevelopmental assessment tool. It is essential to evaluate whether the instrument sufficiently captures the diverse cognitive, behavioral and linguistic competencies of children from mixed ethnic backgrounds. Standardized developmental tools are frequently normed on majority populations, which may inadvertently disadvantage children whose cultural experiences and learning environments differ from those of the dominant group.

Socioeconomic disparities may also underlie the observed differences. Children from mixed ethnic backgrounds may experience reduced access to quality healthcare, early education opportunities or parental support services. Factors such as lower parental education, limited income or inadequate maternal-child health services can indirectly but substantially influence developmental trajectories.

Language exposure within the home environment is another key factor. Bilingual or multilingual households—common in families of mixed ethnicity—may influence patterns of language acquisition and performance on assessments designed to evaluate development in a single dominant language. In such cases, developmental “delays” may instead reflect the cognitive demands of navigating multiple linguistic systems. Moreover, differences in parenting practices and culturally shaped norms around early learning, communication and socialization may impact how children engage with and perform on standardized developmental assessments.

It is essential to avoid generalizations or overly simplified conclusions regarding the category of “mixed ethnicity,” as this group is inherently heterogeneous and encompasses a wide array of cultural, linguistic and socioeconomic contexts. Future research is needed to disentangle the specific characteristics within this broad category that may influence neurodevelopmental outcomes. In particular, evaluating the cultural validity and sensitivity of the assessment tools employed and conducting more granular analyses of ethnicity-related variables would represent important steps toward a more accurate and equitable understanding of child development. Our findings underscore the importance of using culturally adapted assessment tools with demonstrated external validity to minimize potential bias related to language or cultural norms.

Additionally, the finding that children residing in rural areas demonstrated significantly poorer language development, compared to their urban counterparts, points to possible disparities in access to developmental supports and enrichment opportunities. Contributing factors may include reduced availability of early childhood education programs, limited exposure to diverse language models and broader socioeconomic challenges associated with rural living. These results emphasize the need for targeted interventions and continued research to address rural–urban disparities in early childhood development.

Higher paternal education emerged as a significant predictor of more favorable overall developmental scores, consistent with previous literature highlighting the positive influence of parental education—particularly paternal involvement—on children’s cognitive and behavioral outcomes [41].

An unexpected and counterintuitive finding was the observed association between paternal smoking or vaping at the child’s age of five and better developmental outcomes. This result contradicts existing evidence and is likely attributable to a spurious association, residual confounding, or complex indirect relationships not accounted for in the current model [42]. This anomaly should not be interpreted as evidence of a causal relationship and warrants cautious interpretation and further investigation in future research.

A paradoxical result also emerged regarding breastfeeding duration and behavioral outcomes: while children breastfed for 6–12 months (Group B) showed improved behavioral scores, those breastfed for more than 12 months (Group C) did not. One possible explanation for the lack of statistical significance in Group C may be a plateau effect, wherein the benefits of breastfeeding on behavioral development peak within the first year and do not increase further with prolonged duration. Alternatively, the relatively smaller sample sizes in Groups B and C may have limited the statistical power to detect differences.

### Limitations and Future Research

Several limitations of this study must be acknowledged. First, the reliance on self-reported data regarding demographics, environmental factors and breastfeeding practices may introduce recall bias, as parents may overestimate the duration or exclusivity of breastfeeding. However, data collection was conducted under the supervision of a physician or lactation specialist using structured questionnaires designed to minimize such bias.

Second, although the study adjusted for a range of demographic and environmental variables, the possibility of residual confounding cannot be excluded. The moderate R^2^ values observed in the regression models suggest that additional, unmeasured factors may significantly influence developmental outcomes.

Third, the relatively small sample size (n = 92) limits the statistical power and generalizability of the findings. Nonetheless, a post hoc power analysis indicated sufficient power to detect effects in the regression models focused on behavioral development outcomes. Subgroup analyses—particularly for Group B—remain underpowered and findings should be interpreted with caution.

Future studies should aim to recruit larger and more diverse cohorts and incorporate a broader range of measured variables, including detailed evaluations of family environment, parental mental health and levels of cognitive and emotional stimulation in the home. Additionally, it is crucial to assess the validity and cultural appropriateness of developmental assessment tools across diverse ethnic and socioeconomic backgrounds to ensure accurate and equitable evaluation.

Another limitation is the lack of detailed household composition data (e.g., single vs. two-parent households, number of siblings, extended family caregivers) and socioeconomic status, both of which may have influenced developmental outcomes. These factors should be incorporated into future research designs.

Finally, the unequal group sizes—particularly the limited number of participants in the medium breastfeeding duration group (Group B)—may further constrain the statistical power and interpretability of subgroup comparisons. Nevertheless, to reduce the influence of potential the effect of other cofounding factors on neurodevelopment, participants were carefully selected based on strict inclusion and exclusion criteria.

## 5. Conclusions

This study contributes to the growing body of evidence on the role of early nutrition—particularly breastfeeding—in shaping domain-specific aspects of neurodevelopment. Our findings indicate that breastfeeding for 6 to 12 months is associated with improved behavioral outcomes at age five, as demonstrated by multivariate regression analysis. Additionally, group-level differences in behavior and language scores across breastfeeding duration categories suggest potential developmental advantages in these domains. However, these associations were not consistently supported across all analytical approaches. Specifically, partial correlation analyses—adjusted for key demographic and environmental covariates—did not identify a significant independent relationship between breastfeeding duration and behavioral or language outcomes. This discrepancy highlights the likelihood that observed effects may not be driven by breastfeeding alone, but rather by complex interactions between early nutrition and broader contextual factors. These results underscore the need for nuanced interpretation when evaluating the long-term impact of breastfeeding. While breastfeeding may contribute to positive developmental outcomes—particularly in behavioral domains—these effects appear to be context-dependent. Future longitudinal studies employing rigorous methodologies and comprehensive control for confounders are warranted to clarify the durability and specificity of these associations beyond the preschool years.

## Figures and Tables

**Table 1 nutrients-17-02578-t001:** Basic characteristics of the sample.

Characteristics	Group A (n = 64)	Group B (n = 10)	Group C (n = 18)	*p*-Value
**Male, n (%)**	37 (57.8)	5 (50)	11 (61.1)	*p* = 0.849
**Age of the father (m ± SD)**	38.2 ± 5.07	36.9 ± 3.75	39.6 ± 2.94	*p* = 0.137
**Age of the mother (m ± SD)**	35 ± 5.30	32.9 ± 3.41	38.2 ± 4.33	*p* = 0.005 **
**Residence, n (%)**				*p* = 0.893
Urban	59 (93.7)	9 (90)	17 (94.4)	
Rural	4 (6.3)	1 (10)	1 (5.6)	
**Mode of birth, n (%)**				*p* = 0.659
Caesarean section	36 (56.2)	5 (50)	8 (44.4)	
Natural birth	28 (43.8)	5 (50)	10 (55.6)	
**Maternal education, n (%)**				*p* = 0.172
Master	40 (62.5)	8 (80)	15 (83.3)	
Secondary	24 (37.5)	2 (20)	3 (16.7)	
**Paternal education, n (%)**				*p* = 0.033 *
Master	28 (43.8)	6 (60)	14 (77.8)	
Secondary	36 (56.3)	4 (40)	4 (22.2)	
**Maternal smoking during pregnancy, n (%)**	14 (21.9)	0	1 (5.6)	*p* = 0.085
**Parental smoking at birth, n (%)**	35 (54.7)	2 (20)	7 (38.9)	*p* = 0.087
**Type of feeding, n (%)**				*p* < 0.001 ***
Exclusive breastfeeding	4 (6.3)	10 (100)	11 (61.1)	
Mixed	41 (64.1)	0	7 (38.9)	
Formula feeding	19 (29.7)	0	0	

* *p* < 0.05, ** *p* < 0.01, *** *p* < 0.001; Group A: breastfed for 6 months or less, n = 64 participants; Group B: breastfed for between 6 and 12 months, n = 10 participants; Group C: breastfed for more than 12 months, n = 18 participants.

**Table 2 nutrients-17-02578-t002:** Developmental outcomes at 5 years: One-Way ANOVA analysis.

Domain	Group A (≤6 m) Mean (SD)	Group B (6–12 m) Mean (SD)	Group C (>12 m) Mean (SD)	*p*-Value
Motor development	12.23 (18.57)	7.00 (5.16)	8.06 (8.71)	0.195
Language development	12.91 (16.66)	11.60 (21.60)	5.56 (6.98)	**0.037 ***
Articulation	7.05 (13.90)	6.10 (14.15)	6.72 (11.92)	0.98
Nonverbal intelligence	10.08 (17.81)	7.00 (7.82)	8.50 (9.62)	0.66
Behavior	5.67 (8.77)	0.90 (1.91)	3.78 (5.61)	**0.001 ****
Total developmental score	47.94 (60.59)	32.60 (38.84)	32.61 (29.58)	0.313

* *p* < 0.05, ** *p* < 0.01; Group A: breastfed for 6 months or less, N = 64 participants; Group B: breastfed for between 6 and 12 months, N = 10 participants; Group C: breastfed for more than 12 months, N = 18 participants; please note that higher scores indicate poorer neurodevelopment and lower scores indicate better neurodevelopment.

**Table 3 nutrients-17-02578-t003:** Overall model test for total developmental score.

Model	R	R^2^	AIC	BIC	F	*df*1	*df*2	*p*
1	0.522	0.273	977	1007	3.00	10	80	**0.003 ****

** *p* < 0.01.

**Table 4 nutrients-17-02578-t004:** Model coefficients–total developmental score at 5 years.

Predictor	Estimate (SE)	*p*-Value
*Intercept*	−30.47 (46.55)	0.515
*Breastfeeding duration* (6–12 m vs. ≤6 m)	−6.97 (17.42)	0.690
*Breastfeeding duration* (>12 m vs. ≤6 m)	−6.80 (13.84)	0.624
*Father’s education* (secondary vs. higher)	19.50 (11.34)	0.089
*Ethnicity* (mixed vs. Bulgarian)	98.70 (36.94)	**0.009 ****
*Ethnicity* (Roma vs. Bulgarian)	28.93 (22.90)	0.210
*Ethnicity* (Turkish vs. Bulgarian)	−5.03 (49.78)	0.920
*Region of residence* (rural vs. urban)	54.02 (22.04)	**0.016 ***
*Father’s age*	1.64 (1.17)	0.165
*Paternal smoking or vaping when the child was 5 years old* (yes vs. no)	−24.29 (11.92)	**0.045 ***
*Maternal smoking or vaping when the child was 5 years old* (yes vs. no)	10.66 (11.46)	0.355

* *p* < 0.05, ** *p* < 0.01; please note that higher scores indicate poorer neurodevelopment and lower scores indicate better neurodevelopment.

**Table 5 nutrients-17-02578-t005:** Overall model test for behavioral outcomes.

Model	R	R^2^	AIC	BIC	F	*df*1	*df*2	*p*
1	0.545	0.297	631	671	2.29	14	76	**0.011 ****

** *p* < 0.01.

**Table 6 nutrients-17-02578-t006:** Model coefficients–behavioral outcomes at 5 years.

Predictor	Estimate (SE)	*p*-Value
*Intercept*	6.35 (2.39)	**0.010 ****
*Breastfeeding duration* (6–12 m vs. ≤6 m)	−5.88 (2.60)	**0.026 ***
*Breastfeeding duration* (>12 m vs. ≤6 m)	−0.83 (2.09)	0.693
*Father’s education* (secondary vs. higher)	3.58 (1.78)	**0.048 ***
*Ethnicity* (mixed vs. Bulgarian)	12.55 (5.42)	**0.023 ***
*Ethnicity* (Roma vs. Bulgarian)	3.33 (3.48)	0.341
*Ethnicity* (Turkish vs. Bulgarian)	0.66 (7.36)	0.929
*Region of residence* (rural vs. urban)	4.80 (3.27)	0.146
*Paternal smoking or vaping when the child was 5 years old* (yes vs. no)	−3.78 (1.79)	**0.038 ***
*Maternal smoking or vaping when the child was 5 years old* (yes vs. no)	0.50 (1.72)	0.774
*Family history of asthma* (yes vs. no)	3.07 (1.80)	0.093
*Maternal allergy* (yes vs. no)	−3.00 (1.85)	0.108
*Paternal allergy* (yes vs. no)	−2.46 (1.77)	0.170

* *p* < 0.05, ** *p* < 0.01; please note that higher scores indicate poorer neurodevelopment and lower scores indicate better neurodevelopment.

**Table 7 nutrients-17-02578-t007:** Overall model test for language development.

Model	R	R^2^	AIC	BIC	F	*df*1	*df*2	*p*
1	0.570	0.325	757	799	2.40	15	75	**0.007 ****

** *p* < 0.01.

**Table 8 nutrients-17-02578-t008:** Model coefficients–language development at 5 years.

Predictor	Estimate (SE)	*p*-Value
*Intercept*	8.50 (4.28)	**0.051 ***
*Breastfeeding duration* (6–12 m vs. ≤6 m)	3.78 (5.33)	0.481
*Breastfeeding duration* (>12 m vs. ≤6 m)	−2.02 (4.08)	0.621
*Father’s education* (secondary vs. higher)	2.98 (3.80)	0.436
*Mother’s education* (secondary vs. higher)	1.83 (3.98)	0.647
*Ethnicity* (mixed vs. Bulgarian)	22.11 (10.8)	**0.045 ***
*Ethnicity* (Roma vs. Bulgarian)	9.235 (6.93)	0.187
*Ethnicity* (Turkish vs. Bulgarian)	0.912 (14.56)	0.950
*Region of residence* (rural vs. urban)	20.24 (6.83)	**0.004 ****
*Maternal smoking or vaping when the child was 5 years old* (yes vs. no)	4.58 (3.36)	0.177
*Family history of atopic dermatitis* (yes vs. no)	−4.45 (3.52)	0.211
*Maternal allergy* (yes vs. no)	−3.41 (3.55)	0.339
*Child’s history of atopic dermatitis* (yes vs. no)	−2.41 (4.47)	0.592

* *p* < 0.05, ** *p* < 0.01,; please note that higher scores indicate poorer neurodevelopment and lower scores indicate better neurodevelopment.

## Data Availability

The datasets presented in this article are not readily available because they are part of an ongoing study. Requests to access the datasets should be directed to Prof. Dr. R. Pancheva, MD.

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
