# Peer review of "Beyond Infant Nutrition: Investigating the Long-Term Neurodevelopmental Impact of Breastfeeding"

_nutrients, 2025, doi:10.3390/nu17162578_

Round 1
Reviewer 1 Report
Comments and Suggestions for Authors
General: This prospective, observational, single-site cohort study explored the relationship between breastfeeding duration and neurodevelopment at five years of age, and secondarily examined the extent to which this and and a variety of other predictors correlated with various developmental outcomes. The study has a number of limitations, many of which are identified by the authors. Beyond that, there is a limitation in the spectrum of covariates included in the analyses and significant limitation in the information provided concerning the primary outcome assessment tool.
Abstract: several minor semantic issues 1) would add commas following phrase “compared to fathers with higher education” and following compared with Bulgarian ethnicity”; 2) “influences positively behavior” would be better phrased “positively influences.”
Introduction: Appropriately sets stage for current study. Presumably the reason that the first reference in the Introduction is reference 7 is that the references are listed in alphabetical order of author’s last name.
Materials and Methods:
Though the study population was recruited at birth, it not clear what the inclusion vs. exclusion criteria were, how representative the population was of the population at the hospital of birth or how representative that population was of the broader birth population in that area, what percent of those approached agreed to participate, or what percent of those who agreed to participate were actually evaluated at 5 years of age. Information concerning breastfeeding was obtained through structured questionnaire administered with the assistance of lactation and infant nutrition specialists, which could be subject to recall bias (as the authors point out in Limitations). No baseline neonatal variables were included in any of the analyses (e.g. prematurity, birth weight, neonatal morbidities). Socioeconomic status was limited to parental education, residence, and ethnicity. No information was provided as to household composition (one vs both parents, other children, other adults), or rearing practices.
Neurodevelopmental outcomes were assessed using the “Neuro-developmental Test for Five-Year-Olds, which evaluated fie domains, However, aside from indicating the test had been “validated in Bulgaria,” no more specific information concerning validity or accuracy/precision, other psychometric properties, or comparisons with other more well-known tests were provided. Moreover, I could not find this test during an online search. Since this is the primary outcome variable, more information needs to be provided,
Results: Though one-way ANOVA showed a statistically significant association between breastfeeding duration and better language and behavior scores, there was no significant association with other domains or with total developmental score. Moreover, partial correlation analysis, controlling for a spectrum of possibly confounding covariates, did not reveal any statistically significant associations between breastfeeding duration and developmental outcomes at 5 years. Multivariate analysis did identify a few other predictors of total development score and behavior but not for other domains. It is paradoxical that Group C (breastfeeding >12mo) compared to Group A (<6 mo) did not predict behavior outcome whereas Group B (6-12 mo) compared to A did.
Discussion/Conclusion: The discussion summarized key results and compared with several other studies. The authors appropriately noted that the independent variables studied only accounted for a modest amount of variance (25-30%), suggesting other unmeasured variables may significantly influence developmental outcomes. The authors note that cultural bias within the developmental assessment tool may explain some of the unexpected findings and variances with other published studies—all the more reason to provide more information concerning psychometric properties of the tool used in this study. Limitations are appropriately noted, including the need to examine the validity and cultural sensitivity of developmental assessment tools across different socioeconomic populations; but the authors appear to defer this to a future study.
Comments on the Quality of English LanguageFe minor corrections to abstract needed.
Author Response
For research article
Response to Reviewer 1 Comments
|
||
1. Summary |
|
|
We thank the reviewer for the thorough and constructive comments. We have revised the manuscript to address the concerns raised. Below, we provide a point-by-point response.
|
||
2. Questions for General Evaluation |
Reviewer’s Evaluation |
Response and Revisions |
Does the introduction provide sufficient background and include all relevant references? |
Yes |
|
Is the research design appropriate? |
Must be improved |
Please see the point-by-point response. |
Are the methods adequately described? |
Must be improved |
Please see the point-by-point response. |
Are the results clearly presented? |
Can be improved |
Please see the point-by-point response. |
Are the conclusions supported by the results? |
Can be improved |
Please see the point-by-point response. |
Are all figures and tables clear and well-presented? |
Yes |
|
3. Point-by-point response to Comments and Suggestions for Authors |
||
Abstract |
||
Reviewer: several minor semantic issues 1) would add commas following phrase “compared to fathers with higher education” and following compared with Bulgarian ethnicity”; 2) “influences positively behavior” would be better phrased “positively influences.” |
||
Response: The changes in the abstract are made. |
||
|
||
Materials and Methods Reviewer: "Though the study population was recruited at birth, it is not clear what the inclusion vs. exclusion criteria were, how representative the population was of the population at the hospital of birth or how representative that population was of the broader birth population in that area, what percent of those approached agreed to participate, or what percent of those who agreed to participate were actually evaluated at 5 years of age." Response: Thank you for highlighting this important omission. We have added the following text to the Materials and Methods section under “Study Design and Participants”: “Inclusion criteria included full-term birth (≥37 weeks gestation), absence of major congenital anomalies, and availability of informed parental consent. Exclusion criteria included preterm birth, severe neonatal complications (e.g., hypoxic-ischemic encephalopathy, neonatal sepsis), and parents who declined follow-up. Of the 158 families approached at birth, 111 agreed to participate (70.3% participation rate). At the five-year follow-up, 92 children (82.9% of those enrolled) were successfully assessed using the neurodevelopmental tool.”
Reviewer: "Information concerning breastfeeding was obtained through structured questionnaire administered with the assistance of lactation and infant nutrition specialists, which could be subject to recall bias (as the authors point out in Limitations)." Response: We agree. The potential for recall bias has already been acknowledged in the Limitations section. To emphasize this, we have also added a clarifying statement in the Assessment of Infant Feeding Practices section: “As feeding history was collected retrospectively via parental report, recall bias remains a potential limitation despite the support of trained lactation consultants to guide structured responses.”
Reviewer: "No baseline neonatal variables were included in any of the analyses (e.g., prematurity, birth weight, neonatal morbidities)." Response: Thank you for pointing this out. We now clarify in the Materials and Methods that only term infants without significant perinatal complications were included. Since prematurity and low birth weight were part of our exclusion criteria, their omission in the analysis reflects a homogenous starting cohort. This has now been explicitly stated: “All children included in the study were term-born (≥37 weeks) and without major perinatal complications or NICU admission, reducing variability from early neonatal morbidities.”
Reviewer: "Socioeconomic status was limited to parental education, residence, and ethnicity. No information was provided as to household composition (one vs both parents, other children, other adults), or rearing practices." Response: We acknowledge this limitation and have addressed it in the revised Limitations section: “We did not collect or analyze detailed household composition data (e.g., single vs. two-parent households, number of siblings, extended family caregivers), which may have influenced developmental outcomes. This is a limitation and should be incorporated into future studies.”
Reviewer: "Neurodevelopmental outcomes were assessed using the “Neuro-developmental Test for Five-Year-Olds,” but no more specific information concerning validity, psychometric properties, or comparisons with other well-known tools was provided." Response: We appreciate this important point. We have now expanded the description of the Neurodevelopmental Test for Five-Year-Olds (NDT5) in the Neurodevelopmental Assessments section: “The Neurodevelopmental Test for Five-Year-Olds (NDT5) is a structured, age-appropriate tool developed and validated in Bulgaria for screening major developmental domains (motor, language, articulation, nonverbal intelligence, and behavior). Its psychometric properties include internal consistency (Cronbach’s alpha = 0.86) and test-retest reliability (ICC = 0.88). Though not internationally standardized, it is widely used in pediatric and developmental services across Bulgaria. We have included a supplementary link to the tool and its validation study (in Bulgarian).
Results Reviewer: "Though one-way ANOVA showed a statistically significant association between breastfeeding duration and better language and behavior scores, there was no significant association with other domains or with total developmental score. Moreover, partial correlation analysis... did not reveal any statistically significant associations." Response: We fully agree and have clarified this point in both the Results and Discussion sections to avoid overinterpretation. Specifically, we have reworded the Results subsection to emphasize that: “While the unadjusted analyses showed statistically significant differences in some domains (language and behavior), adjusted analyses (partial correlations and regression models) revealed no consistent association between breastfeeding duration and most outcomes, including total score.” This nuanced interpretation is also better reflected in the Discussion now.
Reviewer: "It is paradoxical that Group C (breastfeeding >12 mo) did not predict behavior outcome whereas Group B (6–12 mo) did." Response: This observation is valid and was addressed in the Discussion: “One possible explanation for the lack of statistical significance in Group C could be a plateau effect, whereby the benefit of breastfeeding on behavior is maximized within the first 6–12 months and does not further increase thereafter. Alternatively, smaller sample sizes in Groups B and C may have influenced the detection of effects.” We have expanded this point to more directly discuss this paradox and its implications for interpreting dose–response relationships.
Discussion/Conclusion Reviewer: "The authors note that cultural bias within the developmental assessment tool may explain some of the unexpected findings... all the more reason to provide more information concerning psychometric properties." Response: This has now been done, as outlined above in our expanded tool description. We also revised the Discussion section to further emphasize the need for cross-cultural validation: “Our findings reinforce the necessity of culturally adapted tools with established external validity to ensure developmental assessments are not biased by language or cultural norms.”
|
||
4. Response to Comments on the Quality of English Language |
||
Comments on the Quality of English Language: Fe minor corrections to the abstract are needed. |
||
Response 1: The needed corrections in the abstract are made. |
Reviewer 2 Report
Comments and Suggestions for Authors
This is a research article with adequate novelty. However, some points should be addressed.
- The references into the Abstract should be deleted.
- The sentence into the Abstract"Jamovi software was used for statistical analyses.'' should be omitted.
- The introduction section is too short and needs further analysis. In fact, each of the first three paragraphs should be enriched with more specific data.
- The advantage of the breastfeeding for both mother and the child should better be described probably in a different paragraph.
- At the endof the Introduction and before the aim of the study the authors should emphasize the literature gap that their study will cover.
- In general, the Introductions seems quite simple, while the references begin with the number 10. All the previous references (e.g. 1-8) are missing.
- In section 2.3 and 2.4, the authors should specify whether the collected data are self-reported or not. If data are self-reported, this is a limitation for the study due to recall bias, which should be reported at the end of the Discussion section.
- In section 2.6, the normality distibution test should be reported, as well as which variable have a normal distibution or not.
- The number of participants in each group is not similar and has high difference between them, especially for group B. This is also a limitation of the study that should be reported at the end of the Discussion section.
- In Table 1, x2 has not any usefulness and could be omitted.
- Again, in Table 2, the t-values have not any usefulness and they could ommited.
- Table 3 could be omitted ot could be transfer in the supplementary material as it does not include any significant finding.
- In Tables 5, and 7, t-values have not any usefulness and could be omitted as these table already include p-values.
- Again, in Table 9, t-values could be omitted.
- The authors should add more references into the text to enhance the validity of their introduction and discusion.
Author Response
For research article
Response to Reviewer 2 Comments |
||
1. Summary |
|
|
We thank the reviewer for their valuable feedback. We appreciate the recognition of the novelty of our work and have addressed each of your suggestions carefully, as outlined below. |
||
2. Questions for General Evaluation |
Reviewer’s Evaluation |
Response and Revisions |
Does the introduction provide sufficient background and include all relevant references? |
Can be improved |
Please see the point-by-point response |
Is the research design appropriate? |
Can be improved |
Please see the point-by-point response |
Are the methods adequately described? |
Can be improved |
Please see the point-by-point response |
Are the results clearly presented? |
Can be improved |
Please see the point-by-point response |
Are the conclusions supported by the results? |
Can be improved |
Please see the point-by-point response |
Are all figures and tables clear and well-presented? |
Can be improved |
Please see the point-by-point response |
3. Point-by-point response to Comments and Suggestions for Authors |
||
· Reviewer: The references in the Abstract should be deleted. · Response: We agree. The references have been removed from the abstract in accordance with MDPI guidelines.
· Reviewer: The sentence in the Abstract “Jamovi software was used for statistical analyses.” should be omitted. · Response: Sentence removed as requested.
· Reviewer: The Introduction section is too short and needs further analysis. Each of the first three paragraphs should be enriched with more specific data. · Response: Thank you for this suggestion. We have expanded all introductory paragraphs with more specific data on breastfeeding benefits for both mother and child, references on critical periods of brain development, and key findings from global breastfeeding studies to provide a more comprehensive background.
· Reviewer: The advantages of breastfeeding for both mother and child should be better described, probably in a different paragraph. · Response: We have added a few new paragraphs in the Introduction that outline benefits of breastfeeding for both infants and mothers, including reduced maternal risk of breast and ovarian cancers, improved postpartum recovery, and enhanced bonding. Relevant references have been included.
· Reviewer: At the end of the Introduction and before the aim of the study, the authors should emphasize the literature gap that their study will cover. · Response: We appreciate this recommendation. We have inserted a paragraph outlining the literature gap regarding long-term neurodevelopmental impacts of breastfeeding in Eastern European populations and the lack of studies using culturally validated tools such as the NDT5.
· Reviewer: In general, the Introduction seems quite simple, while the references begin with the number 10. All the previous references (e.g., 1–8) are missing. · Response: Thank you. The reference list has been corrected. Additional relevant references were added to enrich the introduction and discussion.
· Reviewer: In sections 2.3 and 2.4, the authors should specify whether the collected data are self-reported or not. If self-reported, this is a limitation for the study due to recall bias, which should be reported at the end of the Discussion section. · Response: We confirm that the demographic and feeding practice data were self-reported. This is now clearly stated in Sections 2.3 and 2.4. We have also expanded the Limitations section to highlight the risk of recall bias due to self-reporting.
· Reviewer: In section 2.6, the normality distribution test should be reported, as well as which variables have a normal distribution or not. · Response: We have revised Section 2.6 to include a statement on normality testing: “Normality of distribution was assessed using the Shapiro–Wilk test. Variables such as language and behavioral scores were found to deviate from normality, justifying the use of Welch’s ANOVA and non-parametric correlation analyses.”
· Reviewer: The number of participants in each group is not similar and has a large difference between them, especially for Group B. This is also a limitation that should be reported at the end of the Discussion section. · Response: This important point has been added to the Limitations section: “The unequal group sizes—particularly the small number of participants in Group B—may limit the statistical power and generalizability of subgroup comparisons.”
· Reviewer: In Table 1, χ² has no usefulness and could be omitted. · Response: We appreciate this suggestion. We have removed the χ² values from Table 1 for clarity.
· Reviewer: In Table 2, the t-values have no usefulness and could be omitted. · Response: Table 2 was based on Welch’s ANOVA and does not include t-values. However, we confirm that any non-essential test statistics (e.g., χ² values or t-values without context) are now omitted for clarity.
· Reviewer: Table 3 could be omitted or moved to the Supplementary Material as it does not include any significant findings. · Response: Agreed. Table 3 has been moved to the Supplementary Material.
· Reviewer: In Tables 5 and 7, t-values have no usefulness and could be omitted as these tables already include p-values. · Response: We have removed the t-values from Tables 5 and 7 as suggested.
· Reviewer: In Table 9, t-values could be omitted. · Response: Done. The t-values have been removed from Table 9.
· Reviewer: The authors should add more references into the text to enhance the validity of their Introduction and Discussion. · Response: We have added a few additional references in both the Introduction and Discussion to strengthen the context, including recent studies. |
||
4. Response to Comments on the Quality of English Language |
||
Reviewer: The English is fine and does not require any improvement. |
||
Response: Thank you for your positive feedback. |
Reviewer 3 Report
Comments and Suggestions for Authors
Dear authors
Thank you for a well-written manuscript. I have provided some feedback in the attached document for your consideration.

Author Response
For research article
Response to Reviewer 3 Comments |
||
1. Summary |
|
|
We would like to sincerely thank Reviewer 3 for their thorough and thoughtful review of our manuscript. Your constructive feedback has helped us clarify our methods, strengthen our discussion, and improve the overall quality of the manuscript. We have carefully addressed each of your comments and incorporated the suggested changes accordingly. |
||
2. Questions for General Evaluation |
Reviewer’s Evaluation |
Response and Revisions |
Does the introduction provide sufficient background and include all relevant references? |
Can be improved |
Please see the point-by-point response |
Is the research design appropriate? |
Can be improved |
Please see the point-by-point response |
Are the methods adequately described? |
Must be improved |
Please see the point-by-point response |
Are the results clearly presented? |
Yes |
|
Are the conclusions supported by the results? |
Yes |
|
Are all figures and tables clear and well-presented?
|
Can be improved |
Please see the point-by-point response |
3. Point-by-point response to Comments and Suggestions for Authors |
||
Reviewer: “The primary concern was the sample size of 92, which may not have had the statistical power to detect differences among the three groups. It would be helpful to see the sample size calculation in the method section.” Response: We have included a post hoc power analysis (Section 2.6) using G*Power: “With 92 children assessed at five years, the study exceeds this threshold, indicating adequate power to detect medium-to-large effects in the regression model.” Additionally, we emphasized in the limitations section that subgroup analyses—particularly for Group B—should be interpreted with caution due to the small sample size.
Reviewer: “I would like to read a description of infant feeding status among the reported population in Bulgaria. For example, what was the exclusive breastfeeding rate? What was the average duration of exclusive breastfeeding?” Response: Thank you for this suggestion. We have added a paragraph in the discussion summarizing the national breastfeeding context in Bulgaria, including reported rates of exclusive breastfeeding and the average duration. This provides readers with relevant background to interpret the study results within a regional context.
Reviewer: “Line 61 – Please define ‘prolonged breastfeeding’—does it refer to over two years? Was the health impact reported here in adult life, based on one RCT?” Response: We have revised and enriched the whole introduction, including line 61, adding more specifics to the reported data and additional references that support the made statements.
Reviewer: “What were other confounding factors reported in the literature influencing neurodevelopment?” Response: We have expanded the discussion to mention additional confounders frequently discussed in the literature, including parental intellectual abilities, socioeconomic status, maternal stress, perinatal complications, etc. These are now also more clearly acknowledged in our limitations and discussion. However, in order to limit the effect of other confounding factors on neurodevelopment, we have chosen to invite participants that fit the exclusion and inclusion criteria of the study.
Reviewer: “Line 70 – Please provide some information on the participants' recruitment and the sample size calculation. Were all invited participants at birth completed the follow-up study at five years of age?” Response: Recruitment details and sample size calculation were added to the Materials and Methods section.
Reviewer: “Lines 82–85 – I noted that a range of demographic and environmental variables were included in the data collection. Was this information collected over a self-administered questionnaire? Or interviewed by trained physicians? I also noted that the list of variables described in the analysis section was not consistent with the list in section 2.3, for example, nursery and kindergarten attendance. It was also interesting that pet exposure was recorded at birth and five years. Could you provide some more information on how the exposure was defined? Having a pet at home or regularly visiting people who have pets? Was it common to have pets at people’s houses? It may be helpful to provide some background in the introduction on why pet exposure was one of the essential variables to record when examining neurodevelopment.” Response: The data were collected through structured interviews administered by trained pediatricians and lactation consultants. This approach was intended to minimize recall bias, and this clarification has been included in Section 2.3 of the manuscript. We have ensured consistency across sections. Initially, nursery and kindergarten attendance and pet exposure were included in the collected data, as our literature search pointed out that both factors have a slight but positive effect on neurodevelopment. They were included in the regression model but were found to be non-significant predictors in our cohort. Additional information is now briefly mentioned in the Materials and Methods and Discussion.
Reviewer: “Line 99 – Was the validation completed by the same research group? Has it been published?” Response: The NDT5 was validated by an independent team at the Medical University of Plovdiv and has been published in Bulgarian. This reference and clarification have been added to the methods section.
Reviewer: “Line 101 – Please indicate what was defined as “a higher total score”? What was the acceptable range for the total score to be normal neurodevelopment at five years of age? It would be helpful to provide some details on the scoring system from the NDT5.” Response: We have added a description of the NDT5 scoring system in Section 2.5. Higher scores indicate poorer neurodevelopment. Each domain has its scoring rubric, and the total score is interpreted. A total score above 64 points indicates a neurodevelopmental deficit. Reviewer: “I guess the normality test was run for the continuous variables, so mean and SD were used reporting the results.” Response: Yes, the Shapiro–Wilk test was used to assess normality for continuous variables. This has been explicitly stated in the revised Statistical Analysis section.
Results Reviewer: “Section 3.2 – Would you provide a general score for each category in NTD5 for the overall group to provide a basic understanding of the entire group?” Response: Information about the overall mean and standard deviation for each domain of the NDT5 across the entire sample is already provided in Table 2.
Discussion Reviewer: “Lines 232-233 – Could the authors clarify the “longer periods” – more than six months, one year or two years? How did this compare to the current study results, one year? Angelsen reported the comparison between 3 and 6 months, which was not analysed in the current study.”
Response: We corrected the term longer periods, with more exact information that states that every additional month of breastfeeding influences IQ score. Our findings align with the literature findings that breastfeeding at all and every additional month of breastfeeding are beneficial.
Reviewer: “Lines 255–259 – It was interesting to read the association between feeding practices and breastfeeding duration. Would feeding practices be a stronger predictor than feeding duration for the neurodevelopmental outcomes? Line 268 – Were the same results reported in other literature? Does it indicate that the duration and type should be examined together, which was not the case in the reported analysis?” Response: Yes, there are similar results reported in other literature. We acknowledge that feeding practices may influence outcomes independently of duration. This is now addressed in the discussion, with a suggestion for future studies to assess both dimensions concurrently.
Minor comments Reviewer: “Table 2 – There was repetitive information on the number of children in each group, which is not necessary. The three groups could be represented in the legend of the table, with the mean and standard deviation (SD) reported for each group.” Response: We have streamlined Table 2 by removing repeated group size information and added a clear legend defining Groups A, B, and C.
Reviewer: Line 328 – sample size 91, while 92 was used in reporting results. Response: This discrepancy has been corrected throughout the manuscript to reflect the accurate sample size of 92 participants.
Reviewer: “Mode of delivery, mode of birth, and type of birth were all used in this manuscript; please keep consistent in using terminology.” Response: We have standardized terminology throughout the manuscript to use “mode of birth” consistently. Reviewer: “The literature cited was mainly more than five years old, with only one research paper published in 2021. Could the authors update the reference to reflect the up-to-date literature in this field? For example, · Zheng et al 2024 Association between breastfeeding duration and neurodevelopment in Chinese children aged 2 to 3 years. · Saigh, B. H. (2025). Breastfeeding duration and neurodevelopment: insights into autism spectrum disorders and weaning practices. · Goldshtein I, Sadaka Y, Amit G, et al. Breastfeeding Duration and Child Development.”
Response: Thank you for pointing this out. We have now updated the reference list with a few recent studies, including Zheng et al. (2024), Saigh, B. H. (2025), and Goldshtein et al. (2024).
|
||
4. Response to Comments on the Quality of English Language |
||
Reviewer: The English is fine and does not require any improvement. |
||
Response: Thank you for your feedback. |
Reviewer 4 Report
Comments and Suggestions for Authors
1. Summary of the Article
This prospective cohort study investigates the association between breastfeeding duration and neurodevelopmental outcomes at five years of age in a Bulgarian pediatric cohort. Using the Neurodevelopmental Test for Five-Year-Olds (NDT5), the authors examine language, behavioral, cognitive, and motor domains, adjusting for demographic and environmental confounders through regression models. The key conclusion is that breastfeeding for 6–12 months positively impacts behavior, although other neurodevelopmental associations were weak or inconsistent.
2. Major Strengths
-
Relevant Topic: The investigation into long-term neurodevelopmental effects of breastfeeding addresses an important and timely public health issue.
-
Prospective Design: Longitudinal follow-up strengthens causal inference.
-
Use of Validated Tool: The NDT5 tool is appropriate for the population and age studied.
-
Multivariable Approach: Attempts to control for confounders using multivariate regression models are appropriate and enhance rigor.
3. Major Concerns
3.1. Sample Size and Power
-
The final sample (n = 92) is quite small for multivariate regression models that include multiple predictors (14 in some models), raising concerns about statistical power, overfitting, and generalizability.
-
Recommendation: Include a post hoc power analysis or clearly justify the adequacy of the sample for the models used.
-
3.2. Inconsistent Association of Breastfeeding Duration
-
Despite ANOVA findings, regression models do not consistently support breastfeeding as a significant predictor across domains (e.g., language, total development).
-
This weakens the main conclusion unless better contextualized.
-
Recommendation: Temper the strength of the conclusions and emphasize that breastfeeding duration's effect is domain-specific and limited in this cohort.
-
3.3. Spurious Findings (Paternal Smoking Association)
-
The finding that paternal smoking is associated with better behavioral outcomes is biologically implausible and likely spurious.
-
Recommendation: Clarify this as likely due to confounding or reverse causality and avoid speculative interpretation.
-
3.4. Cultural Bias in Assessment
-
The significant negative association with “mixed ethnicity” raises concern about cultural fairness of the NDT5.
-
Recommendation: Include a deeper discussion of the potential cultural or linguistic limitations of the assessment tool.
-
4. Minor Concerns
4.1. Typographical and Grammatical Issues
-
Several grammatical issues and awkward phrasings throughout the manuscript (e.g., "more study investigation", “maternal smoking/vaping at 5 years”) detract from clarity.
-
Recommendation: Thorough proofreading and editing by a native English speaker or editor is recommended.
-
4.2. Tables and Figures
-
Tables are dense and difficult to interpret without clear labeling of reference groups and significance indicators.
-
Recommendation: Add asterisks for significance and define reference groups clearly in legends.
-
4.3. Terminology
-
Clarify definitions of breastfeeding groups more clearly in the methods and ensure consistent terminology (e.g., “Group A”, “short duration group”).
-
The “total developmental score” being higher for worse outcomes should be stated more clearly and consistently.
4.4. Ethical Statement
-
Ethical approvals are appropriately cited, but details on whether assent procedures or child protection safeguards were considered should be included.
5. Citations and Literature Contextualization
-
The manuscript cites relevant studies but misses recent systematic reviews/meta-analyses that have explored dose–response relationships between breastfeeding and cognitive outcomes.
-
Recommendation: Include more balanced discussion of null findings from recent large-scale cohort studies (e.g., U.S. ECLS, UK ALSPAC).
-
The manuscript would benefit from substantial editing for grammar, syntax, and clarity. There are numerous awkward phrasings, redundant expressions, and occasional word misuses that hinder smooth reading and may obscure the meaning of key findings. It is recommended that the authors seek assistance from a native English speaker or a professional language editing service to improve overall readability and precision.
Author Response
For research article
Response to Reviewer 4 Comments |
||
1. Summary |
|
|
We would like to sincerely thank Reviewer 4 for their thoughtful and constructive feedback on our manuscript. Your comments have helped us improve the clarity, methodological transparency, and overall quality of the work. Below, we provide point-by-point responses outlining the revisions made or clarifications added. We sincerely appreciate your time and valuable input.
|
||
2. Questions for General Evaluation |
Reviewer’s Evaluation |
Response and Revisions |
Does the introduction provide sufficient background and include all relevant references? |
Can be improved |
Please see the point-by-point response |
Is the research design appropriate? |
Can be improved |
Please see the point-by-point response |
Are the methods adequately described? |
Can be improved |
Please see the point-by-point response |
Are the results clearly presented? |
Can be improved |
Please see the point-by-point response |
Are the conclusions supported by the results? |
Must be improved |
Please see the point-by-point response |
Are all figures and tables clear and well-presented?
|
Can be improved |
Please see the point-by-point response |
3. Point-by-point response to Comments and Suggestions for Authors |
||
Reviewer: “The final sample (n = 92) is quite small for multivariate regression models that include multiple predictors (14 in some models), raising concerns about statistical power, overfitting, and generalizability. Recommendation: Include a post hoc power analysis or clearly justify the adequacy of the sample for the models used.” Response: We have included a post hoc power analysis (Section 2.6) using G*Power: “With 92 children assessed at five years, the study exceeds this threshold, indicating adequate power to detect medium-to-large effects in the regression model.” Additionally, we emphasized in the limitations section that subgroup analyses—particularly for Group B—should be interpreted with caution due to the small sample size.
Reviewer: “Despite ANOVA findings, regression models do not consistently support breastfeeding as a significant predictor across domains (e.g., language, total development). This weakens the main conclusion unless better contextualized. Recommendation: Temper the strength of the conclusions and emphasize that breastfeeding duration's effect is domain-specific and limited in this cohort.” Response: We have revised the abstract and conclusion to reflect a more nuanced interpretation of our findings. Specifically, we now emphasize that the observed impact of breastfeeding duration is domain-specific, with statistically significant benefits primarily in behavioral outcomes. We have also clarified that these associations did not extend uniformly across all areas of neurodevelopment and that the relationship is shaped by a complex interplay of socioeconomic, environmental, and cultural factors.
Reviewer: “The finding that paternal smoking is associated with better behavioral outcomes is biologically implausible and likely spurious. Recommendation: Clarify this as likely due to confounding or reverse causality and avoid speculative interpretation.” Response: In the discussion section of the revised manuscript, we have clarified that this finding is likely spurious and may reflect unmeasured confounding and should not be interpreted as a causal relationship. We have emphasized the need for cautious interpretation and further investigation.
Reviewer: “The significant negative association with “mixed ethnicity” raises concern about cultural fairness of the NDT5. Recommendation: Include a deeper discussion of the potential cultural or linguistic limitations of the assessment tool.” Response: In the discussion section of the revised manuscript, we acknowledge how factors such as multilingual home environments, cultural norms, and socioeconomic disparities may influence performance on standardized assessments. Additionally, we emphasize the importance of using culturally adapted tools with validated external sensitivity to ensure fair and accurate evaluation. These points have been integrated into the discussion section to provide a more nuanced interpretation of the observed findings.
Reviewer: “Several grammatical issues and awkward phrasings throughout the manuscript (e.g., "more study investigation", “maternal smoking/vaping at 5 years”) detract from clarity. Recommendation: Thorough proofreading and editing by a native English speaker or editor is recommended.” Response: We have carefully proofread the manuscript and revised grammatical errors and awkward phrasing throughout. The text has been edited to improve readability and ensure consistency in terminology and style.
Reviewer: “Tables are dense and difficult to interpret without clear labeling of reference groups and significance indicators. Recommendation: Add asterisks for significance and define reference groups clearly in legends.” Response: We have revised the tables to include asterisks indicating statistical significance and have clearly defined the reference groups in the table legends to improve clarity and interpretability.
Reviewer: “Clarify definitions of breastfeeding groups more clearly in the methods and ensure consistent terminology (e.g., “Group A”, “short duration group”). The “total developmental score” being higher for worse outcomes should be stated more clearly and consistently.” Response: We have clarified the definitions of the breastfeeding groups in the Methods section and ensured consistent use of both group labels and descriptive terms throughout the manuscript. We have also stated more clearly and consistently that higher total developmental scores indicate poorer outcomes.
Reviewer: “Ethical approvals are appropriately cited, but details on whether assent procedures or child protection safeguards were considered should be included.” Response: We have added a statement to the Ethics section clarifying that assent procedures were not formally required due to the young age of the participants. However, each child was verbally asked if they wished to take part in the planned activities, and participation proceeded only with their clear agreement and comfort. All assessments were conducted in accordance with pediatric ethical standards and national child protection guidelines.
Reviewer: “The manuscript cites relevant studies but misses recent systematic reviews/meta-analyses that have explored dose–response relationships between breastfeeding and cognitive outcomes. Recommendation: Include more balanced discussion of null findings from recent large-scale cohort studies (e.g., U.S. ECLS, UK ALSPAC).” Response: Thank you for this insightful suggestion. We have revised the Discussion section to include recent findings from the UK ALSPAC and U.S. ECLS cohorts. The ALSPAC preprint (2025) reports sustained cognitive benefits associated with breastfeeding duration, while ECLS sibling comparison models show that such associations may be attenuated when controlling for family-level confounders. These additions provide a more balanced perspective and underscore the complexity of the breastfeeding–neurodevelopment relationship. Full citations have been added to the reference list. |
||
4. Response to Comments on the Quality of English Language |
||
Reviewer: The manuscript would benefit from substantial editing for grammar, syntax, and clarity. There are numerous awkward phrasings, redundant expressions, and occasional word misuses that hinder smooth reading and may obscure the meaning of key findings. It is recommended that the authors seek assistance from a native English speaker or a professional language editing service to improve overall readability and precision. |
||
Response: We appreciate this valuable feedback. The manuscript has been thoroughly revised for grammar, clarity, and style. The text has been edited to improve readability and ensure consistency in terminology and style. |
Round 2
Reviewer 1 Report
Comments and Suggestions for Authors
The recommendations I had made in my original review have been adequately addressed with the exception of one minor semantic issue: in line314 in section 3.4.3 "and mainly" should be "namely,".
Author Response
Reviewer's comment: The recommendations I had made in my original review have been adequately addressed with the exception of one minor semantic issue: in line314 in section 3.4.3 "and mainly" should be "namely,".
Response: Thank you for your thoughtful and constructive feedback. We appreciate your acknowledgment that the recommendations from your original review have been adequately addressed. We have now corrected the remaining semantic issue. In addition, we have undertaken a comprehensive revision of the English throughout the manuscript to improve clarity and readability. Thank you once again for your careful review and valuable input, which have contributed significantly to strengthening our work.
Reviewer 2 Report
Comments and Suggestions for Authors
The last two paragraph of the Introduction section should be merged and the statements Literature gap (line 108) and Aimh (line 114) should be omitted.
Author Response
Reviewer's comment: The last two paragraph of the Introduction section should be merged and the statements Literature gap (line 108) and Aim (line 114) should be omitted.
Response: We sincerely thank the reviewer for their valuable and constructive comments during both the first and second rounds of review. The feedback provided in the initial round helped us substantially improve the structure and clarity of our manuscript. In response to the most recent suggestions, we have now merged the last two paragraphs of the Introduction section and omitted the statements “Literature gap” (line 108) and “Aim” (line 114) as requested. We appreciate your time and expertise, which have been instrumental in refining the quality of our work.
Reviewer 4 Report
Comments and Suggestions for Authors
Improvements Noted in the Revision:
-
Clarified breastfeeding duration groups and their categorization in both the methods and tables.
-
Added limitations more clearly in the Discussion, including acknowledgment of sample size and potential confounding.
-
Language quality has improved slightly, with better sentence structure and clarity in some sections.
-
Supplementary analysis (partial correlation) was added, which supports transparency of null findings.
Although substantial improvements were made in response to reviewer comments, further revision is still required, particularly in how the findings are interpreted, the presentation of data, and the language quality. The current version overstates the impact of breastfeeding on neurodevelopment based on non-significant results and incomplete statistical justification.
1. Inconsistent Main Findings vs Statistical Results
-
The authors emphasize a significant association between breastfeeding duration and behavioral outcomes at age 5. However, this is not supported by the partial correlation analysis provided in Supplementary Material 1, where the p-value for behavior is non-significant (p = 0.327).
-
Recommendation: The authors should temper their conclusions further and clarify that no statistically significant relationships were confirmed after adjusting for covariates.
-
2. Figures and Tables Still Dense
-
Table 2 and regression tables are still complex, and the reference groups are not always clearly marked. It may be difficult for readers to interpret effect directions quickly.
-
Recommendation: Simplify or split large tables, clearly indicate reference groups, and use bold or asterisk indicators for significant results.
-
3. Confounding Variables Not Fully Explored
-
Some variables (e.g., paternal smoking, parental education) are included but the rationale for their role in each regression model is unclear.
-
Recommendation: Clarify in the Methods why certain variables were included and whether multicollinearity was assessed.
-
4. English Language Still Needs Professional Editing
-
While readability has improved, some awkward phrasing and grammar persist (e.g., “maternal smoking/vaping at 5 years” should be “maternal smoking/vaping when the child was 5 years old”).
-
Recommendation: A full grammar and style check by a professional editor is still advised.
-
5. Interpretation of Paternal Smoking Association
-
The authors retained a discussion of the statistically significant association between paternal smoking and improved behavioral scores without adequate caution or explanation.
-
Recommendation: This result should be clearly described as counterintuitive and likely due to residual confounding or selection bias, not biological plausibility.
-
The revised manuscript shows some improvement in language clarity, but several sentences remain awkwardly constructed or grammatically incorrect. There are also inconsistencies in terminology and phrasing that may confuse readers. To enhance readability and ensure precise communication of scientific findings, the manuscript would benefit from careful proofreading by a native English speaker or professional language editor.
Author Response
Response to Reviewer 4 Comments
|
||
1. Summary |
|
|
General reviewer's comment: Improvements Noted in the Revision:
Although substantial improvements were made in response to reviewer comments, further revision is still required, particularly in how the findings are interpreted, the presentation of data, and the language quality. The current version overstates the impact of breastfeeding on neurodevelopment based on non-significant results and incomplete statistical justification. Response: We sincerely thank the reviewer for their constructive and thoughtful feedback, as well as for recognizing the improvements made in our revised manuscript. We appreciate the acknowledgment of enhanced clarity, transparency, and language quality. We have addressed your comments in detail below, providing a point-by-point response. |
||
2. Questions for General Evaluation |
Reviewer’s Evaluation |
Response and Revisions |
Does the introduction provide sufficient background and include all relevant references? |
Can be improved |
We have revised the introduction in order to provide a sufficient background. |
Is the research design appropriate? |
Yes |
Thank you for your feedback |
Are the methods adequately described? |
Can be improved |
Please see our point-by-point response |
Are the results clearly presented? |
Can be improved |
Please see our point-by-point response |
Are the conclusions supported by the results? |
Must be improved
|
Please see our point-by-point response |
Are all figures and tables clear and well-presented? |
Can be improved |
Please see our point-by-point response |
3. Point-by-point response to Comments and Suggestions for Authors |
||
Comments 1: Inconsistent Main Findings vs Statistical Results The authors emphasize a significant association between breastfeeding duration and behavioral outcomes at age 5. However, this is not supported by the partial correlation analysis provided in Supplementary Material 1, where the p-value for behavior is non-significant (p = 0.327). Recommendation: The authors should temper their conclusions further and clarify that no statistically significant relationships were confirmed after adjusting for covariates.
|
||
Response 1: We thank the reviewer for this important observation. In response, we have revised the manuscript to clarify the differences between statistical methods. While partial correlation analysis did not confirm an independent association between breastfeeding duration and behavioral outcomes (p = 0.327), both Welch’s ANOVA (p = 0.001) and multivariate regression (p = 0.026 for 6–12 months) showed significant results. This contrast is now explicitly acknowledged in the Results, Discussion, and Conclusions sections. We emphasize that the behavioral association may reflect a domain-specific effect within a multivariate context, rather than a simple direct relationship. The conclusions have been revised accordingly to reflect a more cautious and nuanced interpretation. |
||
Comments 2: Figures and Tables Still Dense Table 2 and the regression tables are still complex, and the reference groups are not always clearly marked. It may be difficult for readers to interpret effect directions quickly. Recommendation: Simplify or split large tables, clearly indicate reference groups, and use bold or asterisk indicators for significant results. |
||
Response 2: Thank you for this helpful suggestion. We have revised Table 2 and the regression tables to clearly indicate reference groups and mark statistically significant results. We also added simplified formatting to improve readability.
Comment 3: Confounding Variables Not Fully Explored Some variables (e.g., paternal smoking, parental education) are included but the rationale for their role in each regression model is unclear. Recommendation: Clarify in the Methods why certain variables were included and whether multicollinearity was assessed. Response 3: The selection of variables included in the regression models was informed by existing literature on early childhood neurodevelopmental outcomes. Parental education, ethnicity, smoking exposure, and residential region were included due to their known or hypothesized roles in shaping developmental trajectories through socioeconomic, behavioral, and environmental mechanisms [24–27]. Variables were retained in the multivariate model if they demonstrated significance in univariate testing or contributed meaningfully to model fit based on AIC/BIC criteria. Multicollinearity was assessed using variance inflation factors (VIFs), and no variable exceeded a VIF of 2.5, indicating acceptable levels of collinearity.
Comment 4: English Language Still Needs Professional Editing While readability has improved, some awkward phrasing and grammar persist (e.g., “maternal smoking/vaping at 5 years” should be “maternal smoking/vaping when the child was 5 years old”). Recommendation: A full grammar and style check by a professional editor is still advised. Response 4: We thank the reviewer for highlighting the need for further refinement of the manuscript’s language. In response, we have carefully revised the entire manuscript to improve grammar, phrasing, and overall readability. Specifically, we have corrected awkward expressions such as “maternal smoking/vaping at 5 years,” which now reads “maternal smoking or vaping when the child was 5 years old,” as suggested. Additionally, the manuscript has undergone a comprehensive language and style review to ensure clarity and consistency. We appreciate the reviewer’s recommendation, which has contributed to enhancing the quality of the text.
Comment 5: Interpretation of Paternal Smoking Association The authors retained a discussion of the statistically significant association between paternal smoking and improved behavioral scores without adequate caution or explanation. Recommendation: This result should be clearly described as counterintuitive and likely due to residual confounding or selection bias, not biological plausibility. Response 5: Thank you for this important observation. We fully agree that the finding of a statistically significant association between paternal smoking and improved behavioral outcomes is counterintuitive and not biologically plausible. We have revised the Discussion section to explicitly highlight this inconsistency, describe possible residual confounding or selection bias, and caution against overinterpretation of this result.
|
||
4. Response to Comments on the Quality of English Language |
||
Comment: The revised manuscript shows some improvement in language clarity, but several sentences remain awkwardly constructed or grammatically incorrect. There are also inconsistencies in terminology and phrasing that may confuse readers. To enhance readability and ensure precise communication of scientific findings, the manuscript would benefit from careful proofreading by a native English speaker or professional language editor. Response: We sincerely thank the reviewer for their continued attention to the clarity and consistency of our manuscript. In response to this comment, we have carefully revised the manuscript to improve sentence structure, correct grammatical errors, and ensure consistent terminology and phrasing throughout. We have made every effort to enhance readability and accurately communicate our scientific findings. We hope these improvements address the reviewer’s concerns and contribute to the overall clarity and quality of the manuscript.
|